# Effect of Arbuscular Mycorrhizal Colonization on Cadmium-Mediated Oxidative Stress in *Glycine max* (L.) Merr.

**DOI:** 10.3390/plants9010108

**Published:** 2020-01-15

**Authors:** Alicia S. Molina, Mónica A. Lugo, María V. Pérez Chaca, Silvina Vargas-Gil, Fanny Zirulnik, Jorge Leporati, Nuria Ferrol, Concepción Azcón-Aguilar

**Affiliations:** 1Facultad de Química, Bioquímica y Farmacia (FQByF), Universidad Nacional de San Luis (UNSL), Área Química Biológica, Ejército de los Andes 950, Bloque I, 1er Piso, San Luis, Argentina; perezch@unsl.edu.ar (M.V.P.C.); fzirul@unsl.edu.ar (F.Z.); 2IMIBIO-CONICET-FQByF-UNSL, Área Ecología, Ejército de los Andes 950, Bloque I, 2do Piso, Box 4 San Luis, Argentina; 3Instituto de Patología Vegetal (IPAVE, CIAP-INTA), CONICET, Camino 60 Cuadras Km. 5,5, C.P. 5119 Córdoba, Argentina; vargasgil.silvina@inta.gob.ar; 4Facultas de Ingeniería y Ciencias Agropecuarias (FICA)-UNSL, Área Matemática, Campus Universitario, Ruta 148 Ext. Norte, 2do Piso Ala Este, Villa Mercedes, San Luis, Argentina; jlleporati@unsl.edu.ar; 5Estación Experimental del Zaidín, CSIC, Profesor Albareda 1, 18008 Granada, Spain; nuria.ferrol@eez.csic.es (N.F.); conchi.azcon@eez.csic.es (C.A.-A.)

**Keywords:** glutathione, heavy metal toxicity, mycorrhization, non-protein thiols, plan antioxidant defenses, *Rhizophagus intraradices*, soybean

## Abstract

Cadmium is a heavy metal (HM) that inhibits plant growth and leads to death, causing great losses in yields, especially in Cd hyperaccumulator crops such as *Glycine max* (L.) Merr. (soybean), a worldwide economically important legume. Furthermore, Cd incorporation into the food chain is a health hazard. Oxidative stress (OS) is a plant response to abiotic and biotic stresses with an intracellular burst of reactive oxygen species (ROS) that causes damage to lipids, proteins, and DNA. The arbuscular mycorrhizal fungal (AMF) association is a plant strategy to cope with HM and to alleviate OS. Our aim was to evaluate the mitigation effects of mycorrhization with AMF *Rhizophagus intraradices* on soybean growth, nutrients, Cd accumulation, lipid peroxidation, and the activity of different antioxidant agents under Cd (0.7–1.2 mg kg^−1^ bioavailable Cd) induced OS. Our results suggest that glutathione may act as a signal molecule in a defense response to Cd-induced OS, and mycorrhization may avoid Cd-induced growth inhibition and reduce Cd accumulation in roots. It is discussed that *R. intraradices* mycorrhization would act as a signal, promoting the generation of a soybean cross tolerance response to Cd pollution, therefore evidencing the potential of this AMF association for bioremediation and encouragement of crop development, particularly because it is an interaction between a worldwide cultivated Cd hyperaccumulator plant and an AMF–HM–accumulator commonly present in soils.

## 1. Introduction

The environmental concentration of cadmium (Cd) as well as other heavy metals (HMs) has increased due to anthropogenic activities. Inorganic phosphate fertilizers contain HM traces and represent an important input of Cd to agricultural soils [1]. Cd is not an essential element for plants but can be easily taken up by the roots through plasma membrane transporters of essential metals and transported to the leaves via xylem. The content of Cd in plant tissues can be transferred to the food chain, thereby harming human health [2]. It is well known that Cd produces plant metabolic alterations that lead to a decrease in growth rate, causing a reduction in crop production [3]. Moreover, Cd has been shown to overproduce reactive oxygen species (ROS), usually leading to uncontrolled oxidative stress [4,5]. This effect has been observed in both roots and leaves [2,6].

Reactive oxygen species levels are regulated by the coordinated action of enzymatic and non-enzymatic antioxidant systems. Superoxide dismutase (SOD), catalase (CAT), and peroxidase (APX) participate in the enzymatic antioxidant defense system, whereas glutathione (GSH) is an essential molecule involved in the non-enzymatic antioxidant defense [4]. GSH is synthesized from non-protein thiols (NPTs), and it is a precursor of phytochelatins (PCs) [2,4,5]; these compounds participate in the chelation and the storage of Cd in vacuoles. Different studies suggested a role for GSH in plant redox signaling processes [7,8]. Furthermore, plant adaptation to grow in the presence of Cd was linked to the capacity to maintain a high intracellular GSH concentration [9].

Arbuscular mycorrhizal fungi (AMF) are ubiquitous soil microorganisms that establish a mutualistic symbiotic association with most land plants that is known as arbuscular mycorrhiza (AM). In this association, the fungus assists the plant in uptaking soil low mobility mineral nutrients, especially phosphate, thereby increasing plant nutrition and growth [10,11,12]. The development of this symbiosis also increases plant tolerance to abiotic and biotic stresses, including metal stress [13,14]. AMF can improve plant tolerance to HMs through different mechanisms, e.g., enhancement of nutrient supply or decrease of water stress [15], metal sequestration through the production of binding proteins such as glomalin [16], and bioaccumulation in the inner root cells colonized by fungal structures [14] as well as in the extraradical mycelium [17,18]. Toxic elements such as Cd can travel from soil to roots through AMF hyphae; however, the fungus may also function as a biological barrier against Cd transfer from root tissues to the shoot [19]. Several studies have shown the role of AMF in reducing Cd stress via the attenuation of its negative effects on shoot biomass and the activity of photosystem II, both processes including enzymatic and non-enzymatic antioxidant and defense pathways in *Trifolium repens* and genotypes of *Cajanus cajan* [20,21]. Moreover, AM formed by *Pisum sativum* and *Rhizophagus intraradices* has been proposed as a buffering system association in Cd-induced stress, with differences in the level of the response to Cd stress between pea genotypes and involving the expression and the transcription of genes related to heavy metal chelation pathways as part of the Cd tolerance strategies acting during the symbiosis [22,23].

Soybean [*Glycine max* (L.) Merr.] is an important crop worldwide whose biomass production and nutritional quality are significantly reduced whenever being grown in metal contaminated soils. Previous studies indicate that Cd severely inhibits soybean growth and that both the amount of non-enzymatic antioxidants and the activities of enzymatic antioxidants are altered in response to Cd exposure [9]. Soybean is a well known AMF host that is cultivated in large areas around the world. Therefore, mycorrhizal soybean plants could be employed to remove Cd from metal contaminated soils.

The aim of this work was to evaluate the stress mitigation effect of the AMF *Rhizophagus intraradices* root colonization in soybean plants under Cd-induced oxidative stress. In order to do so, plant growth was examined together with Cd accumulation in plants and soil, plant oxidative stress marker (lipid peroxidation), non-enzymatic defense (GSH and non-protein thiols), and enzymatic defense (ascorbate peroxidase activity). It was hypothesized that soybean mycorrhization with *R. intraradices* can mitigate the effect of Cd-induced stress with the intervention of antioxidant defense processes, and the working hypotheses were:
(a)Cd decreases AMF colonization of soybean roots;(b)The two components of this experimental system (soybean and the *R. intraradices* isolate) behave as efficient accumulators of Cd;(c)The mycorrhization with *R. intraradices* improves the soybean growth under Cd stress; and(d)The symbiotic association with the AMF avoids the increase of lipid peroxidation in soybean caused by Cd oxidative stress.

## 2. Results

### 2.1. Effect of Cd on Soybean Mycorrhizal Colonization

Cadmium addition did not modify AMF colonization in soybean roots at the end of the experiment at the final time (tf) (Table 1). Percentage of root colonization by *R. intraradices* was not modified after 40 days of Cd treatment (tf) with respect to the values of the control mycorrhized (M) plants (91.00% ± 3.59, 85.67% ± 3.09, respectively; *p* = 0.1436) without Cd. Similarly, the addition of Cd did not significantly modify the percentages of arbuscules (72.00% ± 5.14, 70.83% ± 5.02, respectively; *p* = 0.4371) and vesicles (39.33% ± 3.76, 35.67% ± 0.80, respectively; *p* = 0.1918) in the colonized roots (Table 1). Percentage of root colonization, arbuscules, and vesicles also increased significantly from the initial time (t0) to t(f) (Table 1).

### 2.2. Soil and Tissues Cd, P, and Fe Contents in Presence and Absence of AMF after Exposure to Cd

None of the soil element (Cd, P, or Fe) concentrations were significantly different between non-mycorrhyzal (nM) and mycorrhizal (M) soybean plants (Table 2). Bioavailable fractions of Cd (1.20 ± 0.07 ppm), P (39.40 ± 0.80 ppm), and Fe (14.48 ± 0.54 ppm) were determined in the soil used for all treatments at the beginning of the trial.

Soybean roots inoculated with *R. intraradices* took up about half as much Cd (53%) than roots of nM plants. The presence of AMF in soybean roots significantly reduced root Cd concentration (Table 3). The amount of Cd accumulated in both new and old leaves did not differ significantly between M and nM plants. However, considering old and new leaves separately, in the inoculated plants, Cd was more concentrated in the old leaves, while in the nM plants, it was accumulated in the new ones (Table 3).

Under pollution conditions, M plants had a significantly higher P concentration compared to nM soybean roots and old leaves (Table 3). Further, in each treatment, P was significantly more abundant in the roots than in all the leaves (*Leaves) and the old leaves. The accumulation of Fe, an essential micronutrient for the plant, was much higher in the roots than in the leaves of both M and nM plants (Table 3). Furthermore, the presence of mycorrhizal colonization significantly increased root Fe concentration in M plants with respect to nM soybean. The amount of Fe accumulated in leaves (*Leaves) of M and nM plants was not different, but P content was significantly higher in M plants (Table 3).

In the same Cd stress conditions, overall Cd, P, or Fe contents of nM and M soybean tissues after 40 days of addition of Cd were estimated from the roots data plus those of total leaves. These data were averages of six replicates ± SE, and the differences between treatments were analyzed by unpaired t-test. Total Cd content of M soybean (32.06 ± 0.47 ppm) was strikingly lower (53.3%, *p* ≤ 0.0001) than that of nM plants (67.84 ± 0.77 ppm), while the total content of both P and Fe increased (P 19%, *p* = 0.050 and Fe 24%, *p* = 0.026) in soybean plants colonized by AMF (P 6820.80 ± 364.56 ppm and Fe 5390.70 ± 288.71 ppm) compared to non-colonized plants (P 5435.20 ± 68.15 ppm and Fe 4166.70 ± 48.60 ppm).

### 2.3. Growth Parameters of Soybean Plants after Cd Exposure

At the end of Cd treatment (tf), total fresh weight (TFW) of nM+Cd plants was significantly lower than that of nM plants (Figure 1a). By contrast, TFW of M plants was not modified by Cd exposure (Figure 1a). Furthermore, root fresh weight (RFW) of M plants was significantly lower than that of nM (Figure 1b). However, M plants in the presence of Cd (M+Cd) yielded significantly higher RFW than control plants without Cd (M) (Figure 1b). At t0 of Cd addition, stem length (SL) of M plants was not significantly different than SL of nM soybean (Figure 1c). Non-mycorrhizal (nM) plants showed a significant decrease in SL at tf of Cd treatment; however, SL of M plants was not modified after exposure to Cd (M vs M+Cd). Moreover, SL of M soybean plants (with or without Cd) was similar to that added with Cd nM soybean and significantly lower than that of nM (Figure 1c). Comparing nM and M plants, shoot fresh weight (SFW) was equivalent to that at t0 as well as between the untreated controls at the end of the treatment. However, 40 days after treatment, Cd caused a significant decrease in SFW in both nM and inoculated plants (Figure 1d).

### 2.4. Oxidative Damage Response of Soybean Plants to Cd Exposure

The highest level of lipid peroxidation was found in colonized plants at tf. Among colonized soybean plants, M+Cd plants showed a decrease of oxidative damage measured as malon-di-aldehyde (MDA) after 40 days of treatment. Furthermore, lipid oxidative level was significantly higher in M plants than in nM plants; a similar trend was recorded at t0 but without significant differences. Cd treated soybean exhibited similar lipid oxidative level, measured as MDA content, in nM and M plants. Under Cd stress conditions, MDA was not significantly affected in nM plants, whereas it decreased in M plants (Figure 2a).

At t0, GSH content was significantly lower in M than in nM plants. In contrast, at tf, GSH level significantly increased in M plants (Figure 2b). At the end of the experiment, Cd significantly increased GSH level in nM plants (nM vs. nM+Cd). Moreover, in M plants, this high GSH level was similar both in the presence and in the absence of Cd. Furthermore, GSH levels in nM soybean plants were equal at t0 and tf (Figure 2b).

Activity of APX varied non-significantly with AMF colonization at t0 (Figure 2c). However, at tf, Cd exposure and AMF colonization reduced APX activity. The highest inhibition of APX activity was found in M plants exposed to Cd; furthermore, Cd addition of M soybean was more detrimental on APX activity than colonization or Cd addition individually. At tf, M plants had lower APX activity than nM soybean (Figure 2c).

In the absence of Cd, M plants had a significantly higher level of NPTs than nM plants. Cd significantly increased levels of NPTs of nM plants. Conversely, in M plants, the levels NPTs decreased in the presence of Cd. Furthermore, Cd treatment and AMF colonization individually had the same level of NPTs (Figure 2d).

## 3. Discussion

### 3.1. Colonization of Soybean Roots by Rhizophagus Intraradices

The results of this study show that Cd did not alter the AMF colonization of soybean roots, indicating that this metal did not inhibit symbiosis establishment. Although AM fungal colonization usually occurs in highly HM contaminated soils [24], positive (*Zea mays*–*Glomus* sp. isolated from high HM contaminated soils and this work), negative (*Cajanus cajan*–*Funneliformis mosseae*), and neutral effects (*Pisum sativum*–*Rhizophagus intraradices*, *Zea mays*–*Funneliformis mosseae*) of Cd on root colonization have been reported, depending on Cd concentration and plant and fungal species involved in the association [22,23,24,25,26,27].

Although the total Cd concentration supplied to the soil was within the range of a high pollution level (10–50 mg kg^−1^ soil), Cd bioavailability values (0.7–1.2 mg kg^−1^ soil) fell within the values of a slightly polluted soil (1–3 mg kg^−1^ soil) for Argentinian Regulatory Decree 831 of 1993 and National Law 24051 of Hazardous Wastes or the maximum level of an unpolluted soil (0–1 mg kg^−1^ soil) for the European Union and Great Britain Laws. This result may explain the lack of Cd effects on fungal growth and root colonization. Hence, Cd bioavailability rather than total concentration should be considered in order to determine the legal limits of soil contamination relative to AMF colonization effects [18].

### 3.2. Soil and Plant Cd, P, and Fe Contents

No differences in Cd, P, or Fe contents were found between nM and M soybean soils. This result is in agreement with the Cd concentration values in soil reported for *Lonicera japonica* colonized by *Glomus versiforme* and *R. intraradices* [28]. Based on our results, we can only propose some possible explanations for this lack of differences in Cd, P, and Fe soil concentrations between M and nM soybean. On the one hand, both partners of our experimental soybean–*R. intraradices* system are recognized as extremely efficient Cd accumulators [13,29,30] and therefore have the capacity to capture and transport metals. AMF Cd accumulation capacity may have considerable ecological implications such as effects on AMF diversity, forcing possible changes in the richness and the abundance of these fungi by promoting the dominance of the resilient species and the detriment of those with less or without Cd accumulation capacity. Consequently, these possible modifications in the AMF communities could modify plant communities, depending on the degree of association of plant species with the AMF (non-host, facultative mycorrhizal, and mycorrhizal dependent species), which would also depend on the metals accumulated in the soil. The similar soil content of Cd or Fe found in nM and M plants might be attributed to the co-transport/capture/translocation of these metals by soybean from the soil due to their importance as nutrients in plant growth, even in the absence of AMF. On the other hand, the efficiency of metal/HM capture in the plant–AMF interactions varies with AMF isolates (isolates of native AMF collected from contaminated soil are more efficient in the capture of these metals than native isolates from uncontaminated soil), host genetic diversity, and metal content [13,22,27]. Thus, the *R. intraradices* strain we used is native but was not isolated from metal-contaminated soils; this *R. intraradices* isolate likely does not capture metals more efficiently than the soybean variety [*Glycine max* (L.) Merr. cv. Don Mario 4800] used in our experimental system with the metal concentrations applied in this study.

Regarding the soybean capture of P, our results agree with previous reports. Mycorrhizal plants have two Pi uptake pathways: a direct pathway through the epidermal root cells and a mycorrhizal pathway. It has been shown that the contribution of the mycorrhizal pathway ranges from a small percentage to about the entire P of the plant and that, during the symbiosis, the plant changes the strategy of Pi uptake and favors the acquisition of P through the AM pathway over the direct root path [11,31]. The total amount of P obtained through the AM pathway depends on the plant, on the fungal genotypes involved in the symbiosis, and on the concentration of P in the soil solution, and it correlates with the abundance of extraradical hyphae in the soil [32]. Nowadays, it is known that fertilization with Pi increases productivity and stress tolerance and that AMF supplies Pi to the plant, thereby improving not only its nutrition but also its tolerance to multiple stresses such as drought, salinity, and pathogen attack [10,32]. Therefore, under stressful environmental conditions, as in our experimental model of HM contaminated soils, the establishment of the AM symbiosis could increase not only the efficient use of Pi by the plant but also the tolerance to other stresses. Soybean roots inoculated with native *R. intraradices* took up about half of the Cd taken up by roots of nM plants, showing that AMF colonization reduced root Cd concentration, as previously reported [22]. Extra and intraradical mycorrhizal mycelia have been proposed as a potential barrier to the entry of HM into roots through mechanisms such as binding, superficial adsorption, complexation, precipitation, and crystallization of metals on hyphal walls [33]. In particular, AMF extraradical hyphae have been shown to be highly efficient in HM binding, capturing, and translocation in soils [27]. Furthermore, the glomalin released by AMF extraradical hyphae may have been able to chelate Fe, Mn, Pb, and Cd, catching (with low affinity) these HMs to soil surrounding the hyphae [33].

*Rhizophagus intraradices* extraradical mycelia successfully detoxified Cd and another HM such as Cu and Zn [13]. Accordingly, *R. intraradices* tolerance to HM stress has been attributed to the strategy that includes the development of a differential AMF mycelium with extensive growth in less stressed areas [29]. Surprisingly, in our study, *R. intraradices* significantly reduced Cd concentration in soybean roots, whereas root colonization in M plants exposed to Cd pollution did not differ in soil Cd concentrations observed between M and nM soybean soils. Thus, Cd appears to be trapped in the interface between soil and roots, suggesting that the hyphal web of *R. intraradices* in soil could function as a soil–root intermediary, holding Cd inside extraradical hyphae or surrounding Cd by glomalin binding; thus, Cd could never reach the soybean roots despite their high colonization. Furthermore, the vermiculite used in the growth pot culture of soybean has been reported as a substrate that improves the sorption capacity of Al and Fe [34] and HMs such as Cu, Ni, and Cd in water solutions and irrigation systems [35,36]. However, vermiculite has low [35] or middle [34] Cd adsorption capacity compared to other HMs, depending on the pH in the media. Thus, vermiculite adsorption of metal ions increases with increasing pH [34]. Therefore, the low bioavailable Cd values could also be due to the retention of vermiculite in pot substrates.

Considering the total (old and new) leaves, the accumulated Cd did not differ between inoculated and non-inoculated plants. However, when old and new leaves were considered separately, mycorrhized plants concentrated more Cd in old leaves, but nM plants accumulated more Cd in young leaves, as previously reported [37]. Furthermore, in this work, plants did not accumulate Cd in the mycorrhizal roots. Thus, Cd content in leaves does not seem to be related to root colonization by AMF, as previously reported for other Fabaceae [22]. Moreover, it has been shown that Cd accumulation in roots of nM soybean is 10-fold higher than in shoots, with small amounts of Cd being translocated to pods and seeds at low Cd concentration in the soil [30]. In our study, Cd concentration in roots was half in M plants, which could have been due to Cd translocation towards the leaves; however, our results showed a 30-times smaller Cd concentration in the leaves than in the roots, with this ratio being double in nM plants. Therefore, these results evidence that soybean is highly efficient in Cd capture in the roots, where this HM is sequestered, preventing translocation to stem and leaves.

In our experimental model of exposure to Cd, the highest overall concentrations of P and Fe as macro- and micronutrients were found in the roots and the leaves of mycorrhized plants, showing a better nutritional condition for the host plants. This host nutritional improvement through the enhanced nutrient supply by the fungal symbiont is one of the most frequently reported mechanisms through which AMF can improve plant tolerance to HMs [15]. Thus, the soybean–*R. intraradices* plant–AMF model is a successful system to crop in soils with Cd.

### 3.3. Response of Soybean Growth Parameters to Cd Exposure

Considering growth parameters, soybean root colonization by *R. intraradices* negatively affected RFW of plants not treated with Cd. This negative effect could have been due to the energy cost of AMF establishment in the soybean roots also affecting stem growth. In general, AMF have been considered to be predominantly beneficial for plant growth [10]; however, it has been suggested that, when functioning as a parasitic partner, their requirement of carbohydrates from the host plant may depress host growth during the first seedling growth period in non- or facultative-mycorrhizal plants and when hosts grow in soils rich in nutrients [38]. Consequently, the costs of AMF colonization do not have deleterious effects on the host if plant growth is not affected by reduced carbon sources [10]. Therefore, soybean growth reduction by *R. intraradices* may be considered a functional rather than a damaging effect. Surprisingly, soybean TFW was preserved, and M plants were qualitatively and morphologically in better conditions than nM plants (data not shown). Moreover, soybean exposed to Cd also showed an opposite trend in RFW, which decreased in nM and increased in M plants. Thus, mycorrhizal colonization promoted root growth despite Cd exposure. Comparing plant TFW with respect to root and SFW, it is possible that, with Cd exposure, TFW of inoculated plants was the same as that of their controls due to the greater fresh weight of the root. Therefore, the association with *R. intraradices* improved soybean growth under Cd stress, showing the beneficial effects of AMF colonization in plants growing under HM pollution conditions, as previously reported [39].

### 3.4. Oxidative Damage Response to R. intraradices Colonization of Soybean Not Exposed to Cd

Oxidative damage was observed in M soybean plants without Cd addition, with the highest level of MDA/lipid peroxidation being recorded in M plants throughout the bioassay. These results may suggest that colonization per se with *R. intraradices* may produce oxidative damage in soybean. Furthermore, reduced glutathione (GSH) is the most important reducing agent in the non-enzymatic antioxidant defense of plants, and its level indicates the overall plant redox state. In this work, soybean mycorrhization changed GSH content, with low values being recorded with *R. intraradices* colonization at the start of the assay and a significant increase of GSH 40 days later. Thus, GSH content increased in correlation with the decrease of lipid oxidation and MDA levels, which was likely a strategy to afford the plant costs for symbiosis establishment [10,40,41]. Furthermore, there is evidence supporting that the induction/suppression mechanisms associated with plant defense play a key role in AMF colonization and the plant–fungus compatibility in the context of this mutualistic association [40], especially at early stages of AMF–plant interaction. These processes include changes in the permeability of the plasma membrane, activation of plasma membrane-bound enzymes and of kinases, phosphatases, phospholipases, and production of signal molecules, including reactive oxygen species. Regarding abiotic stresses, AMF colonization is also involved in defense mechanisms and processes, possibly with effects on the induction of stress resistance [40]. Moreover, several studies showed that root penetration and colonization by AMF involve a complex cytological and biochemical sequence of events and intracellular changes, including ROS increase and anti-oxidative damaging response [42,43,44].

While a decrease in oxidative damage—measured as MDA after 40 days of treatment—was observed, the activity of APX did not vary significantly with AMF colonization at the start of the assay. The NPTs showed similar patterns of concentration at the start and at the end of treatment. Although AMF colonization per se may produce oxidative burst, M plants are able to react by decreasing lipid oxidation levels in the presence of HM. Moreover, H_2_O_2_ is one of the well-known ROS, functioning as an important signaling compound regarding the interaction of plants and AMF root colonization [10,33,42]. Thus, AMF colonization would regulate oxidative damage [44], improving the defensive response of plants against HM, in agreement with different plant–fungus systems polluted by Cd and other HMs [22].

### 3.5. Oxidative Damage Response to R. intraradices Colonization of Soybean with Cd Exposure

When soybean was cultivated in Cd-polluted soil, APX activity decreased significantly in both M soybean (55%) and nM control (36%), whereas mycorrhization did not show differences between treatments. Our results suggest that, under Cd stress, mycorrhizal soybean regulates oxidative damage, maintaining GSH synthesis higher than APX activity as a possible defense strategy. In the presence of Cd pollution, the lack of differences in oxidative damage measured as MDA between M and nM soybean was also accompanied with a significant increase in the contents of GSH and NPTs in nM soybean. In contrast, soybean colonized by *R. intraradices* maintained GSH levels but reduced the content of NPTs with Cd addition. These results are in agreement with findings reported by Rivera-Becerril et al. [23], who showed that the genes coding for glutamil cisteinil synthetase and GSH synthetase, responsible for GSH synthesis, were activated by Cd in M and nM plants. Here, there was a high increase in NPTs in nM soybean with Cd supply, as expected, because it is well known that Cd promotes the synthesis of these molecules. Instead, M soybean did not react to Cd pollution and presented lower levels of these detoxification compounds than nM plants. Furthermore, these results indicate that, although the trace of HM chelation is involved in soybean responses to Cd stress, plants could not make a great contribution to the Cd tolerance strategies operating in AM symbiosis. In soybean root under Cd stress, mycorrhizal colonization per se might be able to increase NTP levels, suggesting that mycorrhization would not enhance plant defense through NTPs compared to plants growing without symbiosis. Thus, the positive effect of AMF on soybean defensive response through NTPs was reduced in the presence of Cd, suggesting that colonization by *R. intraradices* would not have a protective effect on soybean via NTPs at this Cd level. Even though the frequency of AMF colonization fluctuated under different experimental combinations of plant–AMF–HM as well as with soil types and bioavailability of nutrients, the protective effect of AMF consistently prevailed over the stress caused by HM [33]; our soybean–Cd–*R. intraradices* system is not an exception. The protective effect of AMF against Cd stress has also been observed in a Cd-sensitive genotype of *Funneliformis mosseae* through a differential display of root target protein, e.g., accumulation of ROS when *R. intraradices* colonizes roots of *Medicago truncatula*, *Nicotiana tabacum*, and *Zea mays* [42]. AMF are effective in plant protection against abiotic stress at high HM concentrations [45]. However, many factors such as metal concentration may affect AMF–plant tolerance response to HM during AMF–plant association [39]. When soybean was cultivated in Cd-polluted soil, several biochemical plant responses were in agreement with the first defensive response to mycorrhization and with the secondary buffering response against Cd in mycorrhizal soybean throughout the experimental period. Thus, it is possible that mycorrhizal colonization first predisposes the plant to a reduced defense state but then, once mycorrhization is established, it acts as a buffer to Cd contamination. Therefore, AMF colonization induces stress resistance [40], functioning as a strategy against oxidative stress caused by herbicides [46] or drought [47]. Thus, plants use common pathways and components in the stress biotic–abiotic/response relationship, which are known as cross-tolerance.

Overall, *R. intraradices* colonization in soybean plants without Cd pollution reduced some growth parameters, increased lipid oxidation levels, and produced an initial reduction in GSH levels as an anti-oxidant, suggesting that mycorrhization per se implies an oxidative stress for soybean. These phenomena allow plants to adapt to a range of different stresses after being exposed to one specific stress, with cross-tolerance having a direct and useful application in agriculture [48]. Cross-tolerance occurred by means of the redox signals in the plant cell signaling pathways and the accelerated production of ROS [48,49], which is the most important and common response of plants to various abiotic and biotic stresses (including mycorrhizal colonization). These redox-ROS mechanisms are concomitant with the two major antioxidant plant cell defenses, ascorbate and GSH, as well as the antioxidant enzymes using these antioxidants [49]. 

## 4. Materials and Methods

### 4.1. Biological Material and Experimental Design

The experiment was carried out considering two factors: AMF colonization and Cd exposure. During an initial culture period of 30 days, the soybean plants inoculated with *Rhizophagus intraradices* (Schenck N. C. & G. S. Sm.) Walker C. & A. Schüßler as well as the control plants showed good growth. This period of culture to determine the initial time (t0) was established according to a previous bioassay to evaluate the necessary conditions for mycorrhizal growth. At the 30th day of culture, Cd was added to the substrate for the first time, and this was considered the initial time (t0) of the Cd addition experiment; from t0 and over 40 days (final time = tf), plants were exposed to Cd. Four experimental treatments were assayed, including six replications (one for each plastic pot) per treatment: (1) plants not inoculated with AMF or non-mycorrhizal plants (nM), (2) non-mycorrhizal plants supplied with Cd (nM + Cd), (3) plants inoculated with AMF or mycorrhizal plants (M), and (4) mycorrhizal plants supplied with Cd (M+Cd). Each pot contained 2 soybean plants.

Cultures were established between *Glycine max* (L.) Merr. cv. Don Mario 4800 (hereafter soybean) and an autochthonous species of *R. intraradices* isolated from soil without Cd of the semiarid areas of Córdoba province in Central Argentina. The AMF spores were extracted from 50 g of each soil sample (n = 6), according to Daniels and Skipper [50], and identified under a light microscope using the morphotaxonomic criteria of the International Collection of Vesicular and Arbuscular Mycorrhizal Fungi (http://invam.caf.wvu.edu/). 

Trap cultures using *Sorghum halepense* L. Pers. as host plant and autochthonous *R. intraradices* were established over 6 months to obtain the AMF inocula. The inocula were named root inoculum (consisting of the roots of the trap plants colonized by *R. intraradices*) and soil inoculum (composed of soil remaining after removal of colonized roots, spores, and mycelia of *R. intraradices*). Root samples were taken from trap plants to measure the percentage of AMF colonization using the gridline intersection method [51]. Trap cultures were used as inocula when root colonization by *R. intraradices* reached a minimum of 50%.

Soybean seeds were surface sterilized in a 1% NaClO solution for 30 s, rinsed in distilled water, placed on sterilized filter paper, and incubated at 28 °C over 3 days. Three-day-old seedlings were transferred to plastic pots (two plants per pot) containing 1.25 kg of substrate (1 soil:1 sand:1 vermiculite:1 perlite). The substrate was previously steamed-sterilized twice at 120 °C during 20 min after an interval of 24 h, and its determined physicochemical characteristics were the following: pH (5.83), electrical conductivity (4.3 dSm-1), total organic carbon (6.33%), total nitrogen (0.39%), available phosphorus (39.40 ppm), and organic matter (10.92%). Pots were supplied daily with distilled water to maintain soil moisture close to saturation capacity (e.g., 40% of the volume of the mixture) by keeping a volume and frequency of irrigation of 250 ml water/pot/week. Plants were grown in a greenhouse with a 16 h light/8 darkness photoperiod at 15–30 °C without fertilization.

Three-day-old seedlings were inoculated with *R. intraradices* by adding 1.4 g of root inoculum plus 310 g of soil inoculum per pot; nM plants received 1.4 g of autoclaved root inoculum per pot with the addition of 25 ml of root inoculum washing filtrate to restore the microorganisms associated with soil [22].

Cadmium concentration for irrigation and its final availability to plants were selected by a preliminary test of soil Cd adsorption [52]; Cd distribution coefficient (Kd) in substrate was also calculated [53]. Cadmium concentration remaining in the substrate was determined by inductively coupled plasma optical emission spectrometry (ICP-OES). During Cd treatments, plants received Cd by irrigation with 25 mg CdCl2 L-1 solution; the calculated Cd adsorption was 98.04%, and the calculated final bioavailability was 0.7–1.2 ppm Cd in each pot. Plants (nM and M) were supplied with Cd solution over 40 days [22]. At the end of the experiment, Cd content was measured in the roots and the leaves of nM and M plants.

### 4.2. Evaluation of AMF Colonization

The presence of *R. intraradices* colonization in soybean roots was checked under microscope after root clearing and staining [54]. The percentage of root colonization by *R. intraradices* was calculated under microscope using the intersection method [55].

### 4.3. Determination of Soil and Biomass Element Content

In order to bring the metals into the solution, partial and then total digestion of the soil were conducted using a microwave digester (Microwave System Brand: Milestone, Model: START D, ETHOS lab station with easy wave or easy control software HPR1000/10S, high pressure segmented rotor).

Samples (0.5 g) of dried and sieved (in 2 mm mesh) soil of Cd treatments were treated with two acid digestions. For partial digestion, acids HNO_3_ 65%, HCl 37%, and HF 40% were used with an 8 min 160 °C, 5 min 210 °C, and 20 min 210 °C microwave program. For total digestion, the acids 40% HF and 5% H_3_BO_3_ were used with an 8 min 160 °C and 10 min 160 °C microwave program. Both digestions were performed at microwave power up to 1000 Watt using up to 500 Watt for operations with 3 or fewer vessels simultaneously. Determinations were made directly on the digested samples. A reagent blank was prepared with 5% milli Q water with Nitric Acid.

Soil Cd and P contents were quantified by mass spectrometry using inductively coupled plasma (ICP-MS) with a concentric flow nebulizer (Pneumatic). Iron determinations were made by optical emission spectrometry with inductively coupled plasma (ICP-OES) using a spectrometer (model ICP- 2070 Baird, Bedford, MA, USA). Calibration was carried out following the direct method with certified standards, according to Standard Methods for the Examination of Water and Wastewater (21st edition, 2005).

Bioavailable fractions of Fe and Cd in soil were determined by extraction with DTPA (0.005 M diethylenetriaminepentaacetic acid, 0.1 M triethanolamine, and 0.01 M CaCl_2_) [56]. The available P was determined by the Bray and Kurtz method [57] with an acid extraction mixture (0.025 M HCl and 0.03 M NH_4_F). In the respective extracts obtained, P and Cd available using ICP-MS and Fe available by ICP-OES were analyzed.

The accumulation of Cd, P, and Fe in plants was measured. From the Cd-treated plants, six complete root systems and three groups of old (3rd to 4th) and new (6th to 7th trifoliate pairs) leaves of nM and M plants were extracted and dried at 80 °C to achieve constant weight. Then, 0.1 g of biomass samples was digested with 1 mL of concentrated HNO_3_ in covered polyethylene tubes and placed in an ultrasound bath for 30 seconds; then, 0.5 ml of H_2_O_2_ were added and placed in a thermostatic bath for 1 h. Finally, 5 temperature strokes of 30 s were given at 5 min intervals (reaching the boiling of the samples each time) to reach higher temperatures and achieve the total digestion of the samples. Cadmium, P, and Fe concentration in roots and leaves was determined by mass spectrometry with inductively coupled plasma (ICP-MS) and a concentric flow nebulizer (Pneumatic). A control without biomass was performed following the same procedure.

### 4.4. Growth Parameters

Cd toxicity was evaluated by measuring the total fresh weight (TFW), the root fresh weight (RFW), the shoot fresh weight (SFW), and the stem length (SL). Mean values were obtained from six replicates. Each replicate was a pot with 2 soybean plants.

### 4.5. Measurements of Oxidative Damage and Antioxidant Defenses

Oxidative damage, evaluated as lipid peroxidation levels, and non-enzymatic and enzymatic antioxidant defenses were measured in the leaves of plants in all treatments. Lipid peroxidation was assessed as MDA (malon-di-aldehyde) content [58], and a control value without thiobarbituric acid was determined according to Hodges et al. [59]. MDA was quantified using the molar extinction coefficient of 155 mM^−1^ cm^−1^, and the results were expressed as nmol MDA g^−1^ LFW (leaf fresh weight).

Non-enzymatic antioxidant defense was measured as reduced glutathione content (GSH) according to Akerboom and Sies [60]. The assays were performed using acid extracts of leaves for protein precipitation. The oxidation of GSH was produced by DTNB (5.5′-di-thio-bis-nitrobenzoic acid) to obtain GSSG (oxidized glutathione) and TNB (5-thio-2-nitrobenzoic acid), which were quantified. The homogenate was prepared in liquid air with fresh leaf tissue, 0.1M HCl (10% w/v), and it was centrifuged at 3000× *g* rpm at 4 °C for 20 min. A reaction mixture containing potassium phosphate buffer 0.5 M (pH 7) and the samples were analyzed in tandem with a standard GSH (0.4 mM in 0.1 M HCl) and incubated at 60 °C for 5 min. DTNB (0.100 mL of 1.5 mg/mL in 0.5 M potassium phosphate buffer, pH 7) was added, mixed, and centrifuged at 10,000× *g* rpm for 2 min. The increase in absorbance at 412 nm (TNB) was measured, and the GSH concentration was expressed in moles GSH gr-1 LFW.

To evaluate antioxidant enzyme activity, the total activity of ascorbate peroxidase (APX) was measured as the decrease in the ascorbate absorbance at 290 nm [61]. The mixture reaction contained appropriate dilutions of the samples in 50 mM potassium phosphate buffer (pH 7.4), 0.5 mM ascorbic acid, and 0.1 mM H_2_O_2_. APX activity was expressed as nmol ascorbate min^−1^ mg^−1^ protein in LFW. Protein content was determined according to Bradford [62] using bovine serum albumin as the standard.

Non-protein thiols were measured using soybean homogenates prepared with 0.250 g fresh soybean leaves homogenized in a mortar with 0.1 g of PVP (polyvinyl pyrrolidone) and 2.5 mL of 0.1 N HCl (pH 2). Homogenates were centrifuged at 3000× *g* rpm for 10 min. The reaction mixture containing 0.5 M potassium phosphate buffer (pH 8), 1 mM EDTA (ethylene diamine tetraacetic acid), soybean homogenate, and 1.5 mg/mL DTNB in 0.1N HCl was bath-incubated at 30 °C for 12 to 15 min and centrifuged at 10,000× *g* rpm for 2 min. Absorbance was determined in a spectrophotometer at 412 nm and compared with the absorbance of DTNB (6 mM) as the standard. The amount of non-protein thiols in the sample was expressed as μM of DTNB g^−1^ LFW [63].

### 4.6. Statistical Analysis

In general, mean values were obtained from six replicates over two independent experiments (each replicate was a pool composed by 2 different plants). Prior to statistical analyses, data normality was analyzed by Kolmogorov–Smirnov or Shapiro–Wilks tests, and for the homoscedasticity of variances, Levene and Bartlett tests were used. The percentage of colonization, arbuscules, and vesicles of *R. intraradices* in soybean roots with and without Cd addition as well as Cd, P, and Fe tissue contents were analyzed by a one way ANOVA with Tukey post-test to evaluate the differences between treatments (nM vs. M plants) and tissues (roots vs. all other tissues and old leaves vs. new leaves) in nM and M soybean plants. An unpaired t test was performed only when a pair of means were compared, as it was done with soil Cd, P, and Fe contents. Data on growth parameters and oxidative damage response were analyzed by a two way ANOVA (additive model), with Cd treatment and mycorrhizal inoculation as independent factors. The post-hoc test of the least significant difference (LSD) was used to assess the differences between treatments using a bidirectional ANOVA. Results were considered significantly different at *p* ≤ 0.05. Data were analyzed using the R Project [64] for Statistical Computing 3.6.1 for Windows, free language, 2019. 

## 5. Conclusions

A lower concentration of Cd in the entire plant, a higher concentration of nutrients such as P and Fe, and a better overall reduction status (GSH and NPTs) in mycorrhizal soybean plants are advantages under Cd exposure. However, these advantages were not reflected in the growth parameters evaluated (TFW and SFW), which were similar for M and nM soybean plants. This lack of increased growth in mycorrhizal plants could be related to a detrimental oxidative effect of mycorrhization per se, which is evident in high levels of lipid peroxidation. Therefore, our results suggest that soybean root colonization by *R. intraradices* could be the signal that, as a redox trigger, initiates cross tolerance to Cd addition.

In Argentina, the Cd concentration allowed in agricultural soils is 3 mg Cd kg^−1^ soil (Regulatory Decree 831 of 1993; National Law 24051 of Hazardous Wastes), which is three times the maximum value allowed in the European Union and Great Britain. Under these conditions, the soybean–AMF system evaluated in this work would be a good bioremediation tool to remove Cd from the soil with dual socio-economic and ecological functions.

Considering that the efficiency of metal/HM capture in the plant–AMF interactions varies with AMF isolates, host genetic diversity, and soil metal content, some studies must be developed to elucidate the possible soybean–AMF mechanisms involved in transport, capture, and translocation of soil nutrients. Furthermore, future research on field investigations in agronomic systems should be carried out considering cross tolerance together with studies on the most effective crop–AMF species combinations in different levels of contamination. It will be necessary to use sustainable agronomic systems that counteract the soil HM accumulation, which is mainly generated in large quantities by the use of agrochemical inputs such as phosphate fertilizers.

## Figures and Tables

**Figure 1 plants-09-00108-f001:**
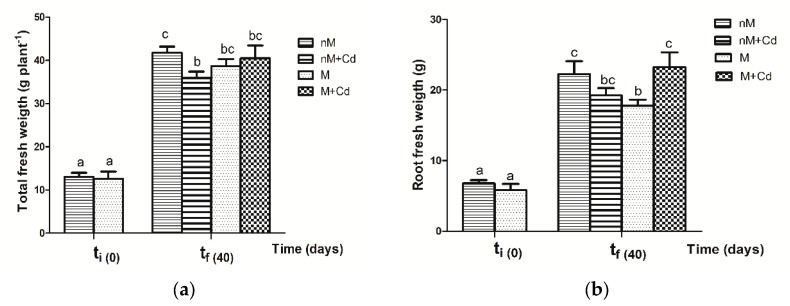
Effect of Cd stress on (**a**) total fresh weight; (**b**) root fresh weight; (**c**) stem length; (**d**) shoot fresh weight of non-mycorrhizal (nM) and mycorrhizal plants (M) inoculated with *Rhizophagus intraradices* at initial time (t0) and after 40 days (tf) of Cd addition. Data are the means of six replicates ± SE for each dependent variable. Different letters indicate significant differences at *p* ≤ 0.05 using a two way ANOVA according to the least significant difference (LSD) post hoc test between treatments.

**Figure 2 plants-09-00108-f002:**
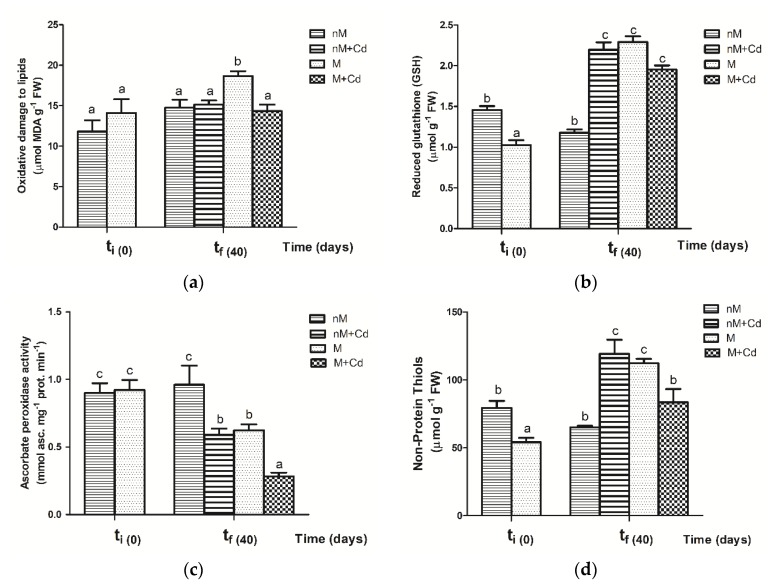
Effect of Cd stress on (**a**) oxidative damage to lipids [malon-di-aldehyde (MDA) content]; (**b**) glutathione (GSH) content; (**c**) peroxidase (APX) activity; (**d**) non-protein thiols (NPTs) content in leaves of non-mycorrhizal (nM) and mycorrhizal (M) soybean plants inoculated with *Rhizophagus intraradices* at initial time (t0) and after 40 days (tf) of Cd exposure. Data are the means of six replicates ± SE for each dependent variable. Different letters indicate significant differences at *p* ≤ 0.05 using a two way ANOVA to the least significant difference (LSD) post hoc test between treatments.

**Table 1 plants-09-00108-t001:** Mycorrhizal colonization of soybean plants inoculated with *Rhizophagus intraradices* supplemented or not with Cadmium (Cd).

Treatment	Harvest Time	Root Colonization (%)	Arbuscules (%)	Vesicles (%)
**Without Cd**	**t0**	66.76 ± 6.21a	36.75 ± 6.21a	1.750 ± 1.10a
	**tf**	85.66 ± 3.09b	70.83 ± 5.02b	35.67 ± 0.80b
**With Cd**	**tf**	91.00 ± 3.59b	72.00 ± 5.1b	39.33±3.76b ^1^

^1^ Data are the means of six replicates ± SE for root colonization, arbuscules, and vesicles. Different letters indicate significant differences (*p* ≤ 0.05); data were analyzed by a one way ANOVA with Tukey post-test between treatments.

**Table 2 plants-09-00108-t002:** Soil Cd, P, and Fe contents (ppm of soil) of non-mycorrhizal and mycorrhizal soybean plants after 40 days of Cd addition.

Soil Elements Concentration (ppm)	Non-Mycorrhizal Soybean + Cd	Mycorrhizal Soybean+ Cd
Cd	10.97 ± 5.06	17.72 ± 8.44
P	529.68 ± 46.3	490.53 ± 49.84
Fe	10375 ± 1742	13582 ± 1441 ^1^

^1^ Data are the means of three replicates ± SE for soil Cd, P, and Fe contents. Average values were not significantly different between mycorrhizal and non-mycorrhizal soybean plants; data were analyzed by an unpaired t test (*p* ≤ 0.05).

**Table 3 plants-09-00108-t003:** Phosphorus, Fe, and Cd concentrations in tissues of non-mycorrhizal (nM) and mycorrhizal (M) soybean plants after 40 days of Cd addition.

Tissues	P (ppm)		Fe (ppm)		Cd (ppm)	
	nM	M	nM	M	nM	M
**Roots**	2176.66 ± 140.18ac	2583.33 ± 59.33be	3884.20 ± 46.83ac	5181.20 ± 160.57be	65.52 ± 0.76ac	30.27 ± 0.50be
**Old leaves**	1187.16 ± 54.87adg	1980.22 ± 204.9bfi	158.56 ± 11.88adg	130.34 ± 29.26afi	0.726 ± 0.020adg	1.056 ± 0.076bfi
**New leaves**	2073.35 ± 15.43ach	2257.14 ± 173.7aei	123.85 ± 10.11adg	79.16 ± 1.15afi	1.600 ± 0.005adh	0.740 ± 0.046bfj
**Leaves ***	1630.25 ± 199.7ad	2118.68 ± 119.95bf	141.20 ± 10.43ad	104.75 ± 17.38af	1.163 ± 0.195ad	0.898 ± 0.081af ^1^

^1^ Data are the means of six replicates ± SE for concentrations of P, Fe, and Cd in soybean tissues. *Leaves are the total values obtained by adding each element concentration in old and new leaves of soybean. Different letters indicate significant differences at *p* ≤ 0.05 by a one way ANOVA with the Tukey post-test. The first letter indicates nM vs. M significance level for each tissue (letters a–b). The second letter represents the significance level of roots vs. all other tissues in nM (letters c–d) and M plants (letters e–f). The third letter shows the significance level of old leaves vs. new leaves in nM (letters g–h) and M plants (letters i–j).

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
