# Peer review of "Effect of Arbuscular Mycorrhizal Colonization on Cadmium-Mediated Oxidative Stress in Glycine max (L.) Merr."

_plants, 2020, doi:10.3390/plants9010108_

Round 1

Reviewer 1 Report

The study reports the effect of AMF on the growth of plants and amounts of cadmium absorbed in the plants. The objectives are clear and the introduction are sound. Materials and methods are appropriate. However, the presentation and statistic analysis are not clearly stated. This can improved much. In table 1, describe clearly the mean separations for each treatment. What data is compared with AMF and Cd at final time. State it clearly. In Table 2, make better way to present the data (ppm) of the the treatments. 

           Concentration of soil elements (ppm) (Mean+/- S.E.) then underline below the title for the treatments.

Means with different letters are significantly different between treatments (not for each treatment).  Available ele. contents ppm (what do you compare? describe clearly or explain why no statics for this.

In table 3, Different letters for the mean separation are all messed up. Authors said unpaired T-test, then the letters are all mixed up. Authors need to check those and make sure. It seems authors do not understand the static analysis right.

In Figure 1 and 2, also make sure to check what treatments authors compare each other and clearly explain these.

Reviewer 2 Report

Abbreviations are not helpful.

Introduction is a great collection of ideas. Please, consider smoothing the flow of ideas even more.´

L96-L97 Did you test the ANOVA assumptions?

L140 Does “at t0as” mean “at t0 as”?

L131-L141 What do TFW and SFW stand for? They should be defined at the first mention.

Results are largely provided based on six-data treatment? Do you six samples are powerful enough to be conclusive?

L182 I think “AM colonization” should be replaced by “AM fungal colonization”.

L198-L199 Could you hypothesize the ecological consequences of Cd accumulation by AM fungi?

L211-L216 What are the drivers of each pathway? Can the anthropogenic disturbance of soybean fields play a role in changing from one pathway to the other?

L218 multiple stresses such as?

L253-L260 Please, keep in mind to add a gap and separate different words.

L284 Why?

Reviewer 3 Report

Dear Authors,

the subject of this study is of considerable interest, especially in this historical period in which the use of agents useful for sustainable management in agriculture are strongly requested.
At the moment, however, this draft is not acceptable for publication. Here are my considerations:
Abstract: line 23: insert the scientific name. Information on the section of materials and methods used is missing. Which inoculum was used? What concentrations of Cd? Please also include some results of interest.
Introduction: Lines 69-72: on which species? Please expand the state of the art on this topic. The same also applies to Lines 73-79. Since it represents the topic of your work, I suggest that you present the hypotheses of your work in the best possible way.
Results: Line 88: insert (Table 1). Since the M&M section is next, please enter the extended forms first and then the abbreviations. Otherwise the reader remains disoriented. For example, Line 89: Mplants? Please first specify that they are mycorrhized. In the M&M and statistical analysis section, you have indicated a series of correct statistical approaches, but in the representation of the results in the table and figures, these statistical analyzes are not indicated. In fact, it is not clear to me how the data was analyzed in Table 1, 2 and 3. I suggest also indicating the significance of the ANOVA. I think it's just a problem of organizing the tables. Why were no statistical analyzes done between fli available elements? If there are no differences it is more correct not to insert the letters of the ANOVA. Below the tables the levels of significance and / or non-significance must be indicated. Line 103-105: they are not needed, they are already in M&M.
In presenting the results, I suggest that you list your results in order with respect to how they are shown in the tables and figures. At the moment reading this information is not completely clear. Line 117-129: please enter a more accurate description of your results. Line 147: what is meant by "before (t0)"? How did you make measurements before time 0? Are there tests before the start of the test? This aspect is not clearly expressed in M&M. The same also applies to the results presented in t0 in figure 2b and d. How is it possible that at t0 there are significant differences between the two treatments? If the test hasn't started yet? Please explain your experimental plan in M&M better.
Discussion: Line 180: Rhizophagus intraradices does not go in italics. Line 184: these effects have been studied in which species. Please extend this discussion so that you can better understand the results you have achieved. Line 188: who says it? Line 229: another HM? Explain better. Line 261: R. intraradices in italics. In the text there are some editing errors, please check. Line 269: insert some examples related to depression. Line 278: increase the discussion with respect to similar jobs. This also applies to Lines 286-290 and 291-299. In these cases the results already presented are repeated, while the discussion is missing. Line 327: different host species? Please explain better. Lines 330-356: in this part the previously presented concepts are repeated. It could be eliminated and some of its parts included in previous discussions, while the main conclusions could be part of the Conclusion section.
M&M: paragraph 4.1 must be reordered, dividing the part relating to the production of the AMF crops and the experiment. Furthermore, it is not clear to me if the 30 days of culture period end with the t0 of the test. Please rewrite this part in a more orderly and chronological way. Specify even better the number of repetitions used. Which experimental plan have you adopted? Are there 6 repetitions? That is, six plants per treatment in total? Line 371: the soil sample where they come from? In paragraph 4.4 I suggest specifying on how many plants the surveys were made, how were they done? Line 445: have they been measured on all the leaves? Statistical analysis: as already indicated, your statistical analyzes do not emerge in the text. Please take inspiration from similar works to present your data more clearly. If you have conducted two-way ANOVAs, I also suggest specifying the significance of the effects due to the CD or the MFA.
Conclusion: Line 485-492: I suggest entering only the most important results and future prospects.

Round 2

Reviewer 3 Report

Dear Authors,

the quality of the paper was improved.

I just have a request for Table 3. It is not clear to me how do you analyse these data. Le letters from one way ANOVA are not clearly presented. Please check for mistake.

Line 146-147: not necessary. This information is already in M&M.
